# Diverse organic molecules on Mars revealed by the first SAM TMAH experiment

Amy J. Williams [1] ✉, Jennifer L. Eigenbrode [2], Maëva Millan[2,3,4], Ross H. Williams[2,5,6], Ophélie M. Mcintosh[3], Samuel Teinturier[2,6,7], Janelle Roach[1], Charles Malespin[2], Amy C. McAdam[2], Paul Mahaffy[2], Alexander B. Bryk[8], Arnaud Buch [9], David Boulesteix [9], Luoth Chou[2], Jason P. Dworkin [2], Valerie Fox[10], Heather B. Franz [2], Caroline Freissinet [3], Daniel P. Glavin [2], Christopher H. House [11], Sarah Stewart Johnson[4], James M. T. Lewis[2,5,12], Angel Mojarro [2,13], Rafael Navarro-Gonzalez [14], Chad Pozarycki[2,15], Andrew Steele [16], Roger E. Summons [13], Cyril Szopa[3], Michael T. Thorpe [2,5,17] & Ashwin R. Vasavada[18]

The search for organic matter on Mars has rapidly evolved in the past decade with simple aromatic, S-heterocycles, and aliphatic organic molecules detected in Gale crater. We report the in situ detection of >20 organic molecules from clay-bearing sandstones in the ~3.5-billion-year-old Knockfarrill Hill member of Glen Torridon, Gale crater, by the Sample Analysis at Mars instrument suite onboard the Curiosity rover. These molecules were liberated by the onboard tetramethylammonium hydroxide wet chemistry experiment. Diverse thermochemolysis products, including benzothiophene, methyl benzoate, and single and dicyclic aromatic molecules were released and detected by evolved gas analysis and gas chromatography-mass spectrometry. Results indicate the experiment successfully released molecules preserved in ancient macromolecular or free organic matter within Martian bedrock despite ~3.5 billion years of diagenesis and radiation exposure.

The ongoing characterization of organic matter on Mars is a pillar of modern robotic exploration, as space agencies send rovers and landers to explore Mars' past and present habitability and to search for signs of life[1]. Within 10 years, we have advanced from the search for organic molecules to identifying indigenous Martian organics. We are now poised to address the source of these organics, whether exogenous (e.g., meteoritic, cometary, or interplanetary dust particles) or endogenous (e.g., abiotically or biologically produced).

[1]Department of Geological Sciences, University of Florida, Gainesville, FL, USA. [2]Solar System Exploration Division, NASA Goddard Space Flight Center, Greenbelt, MD, USA. [3]Laboratoire Atmosphère et Observations Spatiales (LATMOS), LATMOS/IPSL, UVSQ Université Paris-Saclay, Sorbonne Université, CNRS, Guyancourt, France. [4]Department of Biology/STIA, Georgetown University, Washington, DC, USA. [5]Department of Civil and Environmental Engineering and Earth Sciences, University of Notre Dame, Notre Dame, IN, USA. [6]Center for Research and Exploration in Space Science and Technology (CRESST) II, NASA/GSFC, Greenbelt, MD, USA. [7]Department of Physics, Catholic University of America, Washington, DC, USA. [8]Department of Earth and Planetary Science, University of California, Berkeley, Berkeley, CA, USA. [9]Laboratoire Génie des Procédés et Matériaux, CentraleSupélec, University Paris-Saclay, Gif-sur-Yvette, France. [10]Department of Earth Sciences, University of Minnesota, Minneapolis, MN, USA. [11]Department of Geosciences and Earth and Environment Systems Institute, The Pennsylvania State University, University Park, PA, USA. [12]Department of Physics and Astronomy, Howard University, Washington, DC, USA. [13]Department of Earth, Atmospheric and Planetary Sciences, Massachussetts Institute of Technology, Cambridge, MA, USA. [14]Instituto de Ciencias Nucleares, Universidad Nacional Autónoma de México, Ciudad Universitaria, México City, Mexico. [15]School of Chemistry and Biochemistry, Georgia Institute of Technology, Atlanta, GA, USA. [16]Earth and Planetary Laboratory, Carnegie Institution for Science, Washington, DC, USA. [17]Department of Astronomy, University of Maryland, College Park, MD, USA. [18]Jet Propulsion Laboratory, California Institute of Technology, Pasadena, CA, USA. ✉e-mail: amywilliams1@ufl.edu

Previous reports of Martian organics in Gale crater from the Sample Analysis at Mars (SAM) flight model (FM) instrument on the NASA Mars Science Laboratory (MSL) mission's Curiosity rover[2] include chlorobenzene, dichlorobenzene isomers, and $C_2$ to $C_4$ dichloro- and trichloroalkanes detected in the Sheepbed mudstone[3,4]; macromolecular organics including S-bearing components (e.g., thiophene, methyl- and dimethyl-thiophene, dithiapentane, dithiolane) in the Murray formation mudstones of Pahrump Hills and Sheepbed mudstone[5] and in the clay-rich strata of the Glen Torridon region[6]; MTBSTFA (N-methyl-N-(tert-butyldimethylsilyl)trifluoroacetamide)-derivatized benzoic acid (as benzoic acid t-BDMS)[7]; and $C_{10}$ to $C_{12}$ alkanes, proposed to derive from decarboxylated fatty acids[8]. This work reports on SAM's tetramethylammonium hydroxide (TMAH) wet chemistry experiment, and the variety of organic molecules liberated from a clay-rich sandstone in the Glen Torridon region of Gale crater. The molecules are products of the first thermochemolysis experiment performed in situ on a planetary body. These data provide critical insights to optimize the second TMAH experiment on SAM and future TMAH experiments onboard the MOMA instrument on the Rosalind Franklin Mars rover[9] and planned for the DrAMS instrument onboard the Dragonfly mission to Titan[10]. This experiment utilized the strongly alkaline chemical reagent TMAH (25% in methanol), which hydrolyzed sample organics, either free or bound to mineral surfaces, in addition to thermal cleavage (pyrolysis at a maximum of 550 °C) and methylation, releasing organic fragments from free or macromolecular materials as volatile products amenable to gas chromatography-mass spectrometry (GC-MS) analysis.

The lower strata of Aeolis Mons record environments impacted by a variety of diagenetic regimes throughout the Glen Torridon region. These clay-rich deposits that include the Knockfarrill Hill member represent lacustrine facies, with cross-bedded sandstones that indicate a shift towards more energetic fluvial environments[11,12]. The clay mineralogy is dominated primarily by dioctahedral smectites[13] and Fe-rich dioctahedral clays[14]. As in the mudstones at the base of Aeolis Mons, these clay-rich environments, and smectite clay especially, are expected to present an optimal environment for concentrating and preserving organic matter[15,16].

Curiosity drilled and analyzed the Mary Anning 3 (MA3) target on sol 2879. Volatile products were split into two measurements: evolved gas analysis (EGA) and GC-MS. EGA continuously and directly measured bulk gas composition during heating with the mass spectrometer. For GC-MS analysis, evolved gas was periodically subsampled by collection on the hydrocarbon trap (HC). Trapped analytes were then thermally released from the trap and the flow split to two GC columns (GC1 and GC2), run consecutively.

## Results and discussion

The TMAH experimental cup consisted of two sealed foil caps. The outer foil contained the TMAH in methanol, 34 nmol of 1-fluoronaphthalene, and 25 nmol of pyrene, both as recovery standards that would not react under thermochemolysis. The inner foil contained 13 nmol of nonanoic acid as the internal standard. The 1-fluoronaphthalene was detected in both GC1 and GC2 data. The detection of 1-fluoronaphthalene and trimethylamine (TMA) confirms that the cup was punctured and the reagent heated such that the TMAH decomposed to its primary byproducts and the recovery standard was released. The nonanoic acid internal standard should have been methylated by TMAH during thermochemolysis and eluted in GC1 at ~17.5 min and in GC2 at ~15.0 min. However, neither nonanoic acid nor its methyl ester was detected. SAM flight engineering data indicated puncture of the foils containing TMAH and the nonanoic acid. The lack of internal standard detection is therefore an artifact of the experimental design, as benchtop experiments confirm that the sampling cadence (the venting during EGA subsampling), driven by SAM valve configurations, led to the loss of the nonanoic acid. The

pyrene recovery standard elutes outside of the GC1 heating window and GC2 stopped heating prematurely, before possible pyrene release to the detector; thus, the experiment did not enable pyrene detection.

SAM-FM EGA results revealed high-molecular-weight (HMW) molecules with ions having mass-to-charge ratios (m/z) up to 537 (Fig. 1A) and a complex signal due to detector saturation with ions such as TMA (m/z 58), which is produced by the decomposition of TMAH. The sawtooth pattern is due to the subsampling process described above. EGA results for select m/z values are consistent with the presence of benzene, toluene, trimethyl- and tetramethylbenzene, naphthalene, and methylnaphthalene (Table S-1). As the gases from EGA do not interact with the HC trap, these m/z values are not related to any HC trap contamination (Fig. S-28). HMW molecules in both EGA and GC-MS signals (Fig. 1A–C) were detected as the total ion current generated by the sum of ion intervals (bands 15-24) over the m/z 282-537 range. Molecules were identified based both on corroborating National Institute of Standards and Technology (NIST) mass spectra[17] comparisons and, when possible, with retention time (Rt) experiments using the SAM-like GC-MS breadboard (BB) with spare flight columns (i.e.,[18,19]).

Three GC-MS experiments were performed on the Mary Anning sample: neat pyrolysis (heating under He flow without additional reagents), TMAH thermochemolysis, and MTBSTFA derivatization. Seven molecules were detected in the SAM-FM TMAH GC-MS data for MA3 that were absent in the pre-sample-analysis and post-analysis clean-up (Fig. 2). These molecules are trimethylbenzene, tetramethylbenzene, methyl benzoate (benzoic acid methyl ester), dihydronaphthalene, naphthalene, benzothiophene, and methylnaphthalene (Table 1). These 7 molecules represent only a portion of the organics generated with this experiment, with 30 chromatographic peaks associated with one or more molecules. The identity of these remaining molecules cannot be confirmed with retention time analyzes, but we can speculate on their identity based on mass spectra. Of these peaks, only bisilylated water (BSW) is also present in the pre-sample-analysis and post-analysis clean-up (Fig. 2A, C).

MTBSTFA and its reaction products are present within the SAM-SMS. The presence of BSW, among other products, is due to a MTBSTFA cup leak detected since Curiosity's landing[20] and two subsequent full-cup MTBSTFA derivatization experiments (on sols 1909 and 2885)[6,7]. A second known SAM-internal source of organic molecules in GC-MS includes thermal degradation products (primarily benzene and toluene) of Tenax TA adsorbent in the hydrocarbon trap generated every heating cycle used to release trapped sample analytes[21]. MTBSTFA-leak and trap derived molecules can complicate interpretations to discern the source of organics detected in SAM, however those byproducts are well characterized[21].

## Organics detected in GC1

Four molecules detected in GC1 (Table 1) are confirmed as trimethylbenzene, tetramethylbenzene, naphthalene, and benzothiophene (peaks 1, 2, 7, and 8, respectively, Fig. 2B). Mass spectra corresponding to two additional molecules were identified in GC1 data, but laboratory retention time experiments on a SAM GC-MS BB did not confirm these identifications. Regardless, the compelling nature of the mass spectra leads us to report on them here as plausible detections of molecules with very similar fragmentation patterns. These detections include a benzene ring with N- and/or O-bearing functional groups (Peak 3) and a benzene ring with N-, COOH-, and/or $CH_3$- functional groups (Peak 4).

At ~25 min Rt and a column temperature of ~220 °C, a complex co-elution occurs in GC1 (Fig. 1B) which may be due to a column saturation effect. The high flux of BSW and TMA is expected to lead to a column saturation effect that does not permit the full portion of these compounds to efficiently adsorb into the column film. These compounds are initially trapped but then elute at high temperature along with other HMW molecules and the expected column bleed.

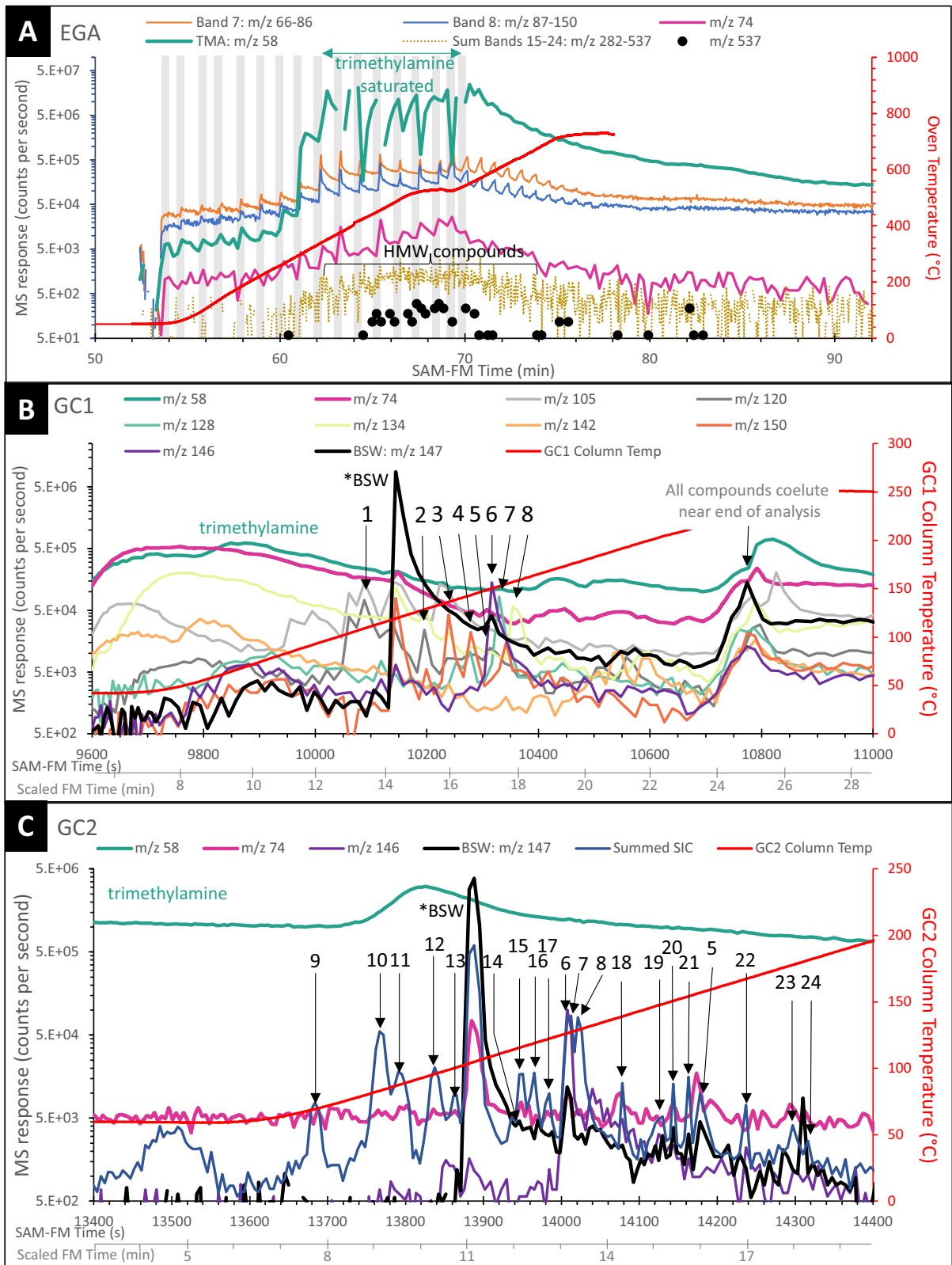

## Organics detected in GC2

More than 20 discrete peaks were present in the GC2 chromatogram (Table 1). Confirmed molecules include methyl benzoate, dihydronaphthalene, naphthalene, benzothiophene, and methylnaphthalene (peaks 11, 15, 7, 8, and 21, respectively).

Sixteen peaks remain unidentified, but similarities in mass spectral fragments can yield insights into likely molecular identities[22].

Identifications include a methylated benzene ring with an amine functional group (Peaks 9, 10, and 17), a multiply-methylated benzene ring with alcohol or methoxy functional groups (Peak 12), a benzene ring with methyl, ethyl, and/or isopropyl groups (Peak 13), a single ring aromatic with methyl and/or ethyl functional groups (Peak 14), a single ring aromatic with methoxy, alcohol, isopropyl and/or methyl functional groups (Peak 16), a single ring aromatic with alcohol, amine, and/

**Fig. 1 | Representative traces extracted from the EGA analysis and chromatograms from the Gas Chromatograph 1 and Gas Chromatograph 2 columns from selected *m/z* values or bands covering a range of masses.** Gray boxes in **A** denote the range of time and oven temperature over which volatiles detected in EGA were subsampled for GC-MS analysis. *m/z* = 74, a trace mass for FAMEs, is included. Eight molecules are highlighted in GC1 in **B**, and 20 molecules are highlighted in GC2 in **C**. For GC2, summed SIC is the sum of selected ions *m/z* 121, 128, 130, 134, 135, 136, 140, 142, 144, 148, 149, 150, 161. Peak 1 – trimethylbenzene, Peak 2 – tetramethylbenzene, Peak 5 – derivatized dimethylsilanediol, Peak 6 - the recovery standard 1-fluoronaphthalene, Peak 7 – naphthalene, Peak 8 – benzothiophene,

Peak 11 methyl benzoate, Peak 15 – dihydronaphthalene, Peak 21 – methylnaphthalene, Peak 23 - diphenylmethane. Peaks 3, 4, 9, 10, 12, 13, 14, 16, 17, 18, 19, 20, 22, and 24 are unidentified. Peak numbers listed correspond to peak identifications in Tables 1 and S1. SAM-FM Time in seconds reflects the serial data collection on GC1 and GC2. Scaled FM Time in minutes reflects the retention time scaled to each GC channel and corresponds to the SAM GC Breadboard retention time in Table S-1. EGA evolved gas analysis, GC gas chromatograph, MS mass spectrometry, FAMEs fatty acid methyl esters, SIC select ion chromatogram, SAM-FM Sample Analysis at Mars Flight Model, BSW bisilylated water, HMW high molecular weight, TMA trimethylamine.

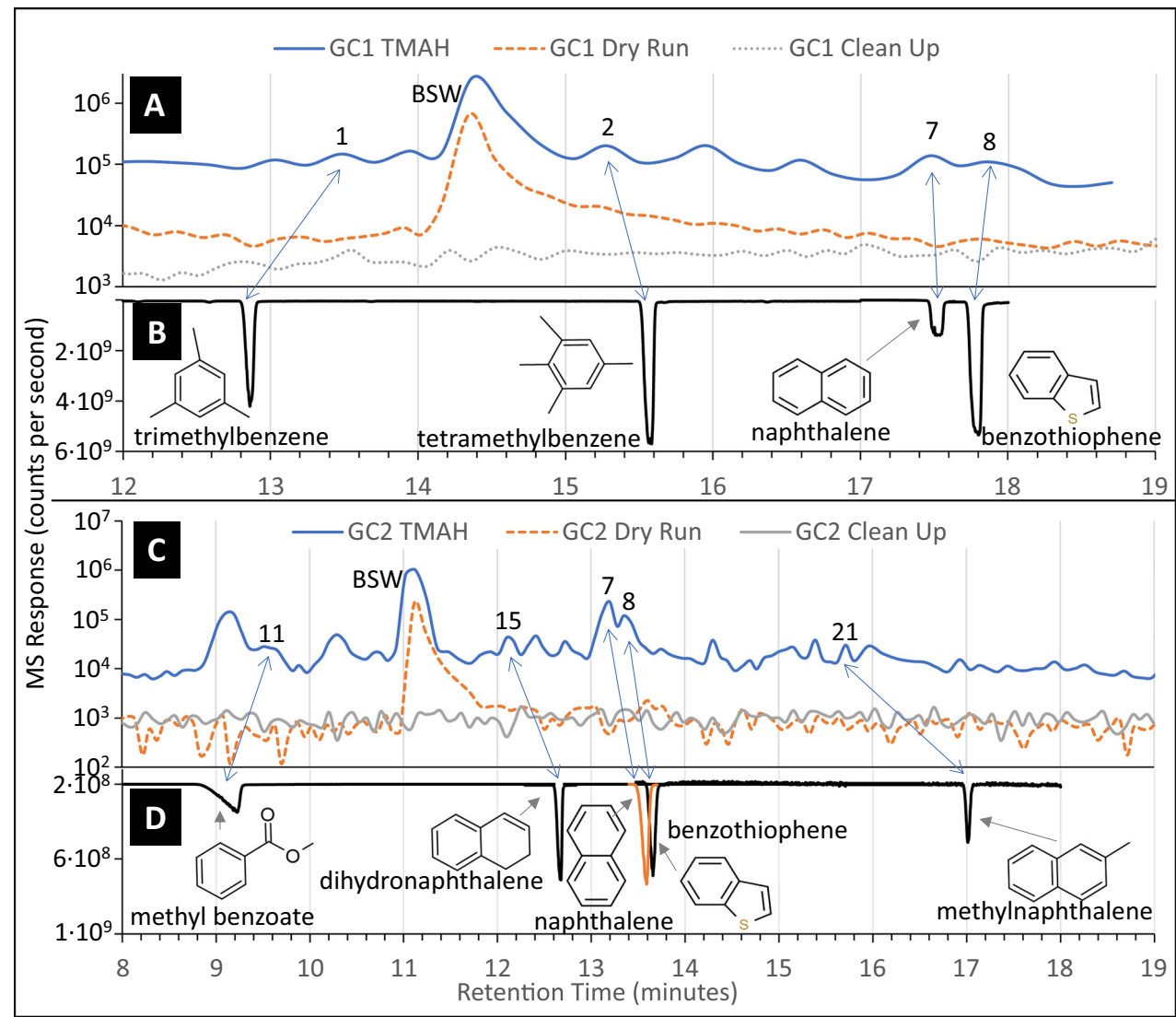

**Fig. 2 | SAM-Flight Model identification of aromatic molecules compared with results from laboratory benchtop retention time experiments.** SAM-Flight Model GC1 and GC2 chromatograms (**A**, **C**) and laboratory benchtop retention time experiments (**B**, **D**) on the SAM-Gas Chromatograph Breadboard confirm the presence of trimethylbenzene, tetramethylbenzene, methyl benzoate, dihydronaphthalene, naphthalene, and benzothiophene in the Mary Anning TMAH experiment. Methylnaphthalene is tentatively identified. Lines represent spectra summed from *m/z* 120, 121, 128, 130, 134, 135, 136, 140, 142, 144, 148, 149, and 150. In

**A**, **C**, solid blue line is the TMAH experiment, dashed orange line is the pre-sample-analysis performed prior to the TMAH experiment (same conditions as the experiment but without a sample or TMAH), dotted gray line is the post-TMAH experiment clean up analysis. In **B**, **D**, black and orange lines represent molecular standards analyzed on the relevant SAM-GC Breadboard GC columns. Numbers above peaks correspond to peak numbers in Fig. 1 and Table 1. GC gas chromatograph, TMAH tetramethylammonium hydroxide, BSW bisilylated water.

or methyl functional groups (Peak 18), a dicyclic aromatic with a methyl group (Peak 19), a multiply-methylated benzene ring (Peak 20), and a methoxy-bearing dicyclic aromatic (Peak 24). Peak 22 is characterized by ions with *m/z* 145, 144, and 146. Comparisons with the NIST library yielded a best match with dimethyl-indole, with the

spectral comparison especially compelling (Fig. S-24). However, the retention time of dimethyl-indole on the SAM-like GC-MS BB GC2 column was offset at 20.7 min relative to the candidate molecule at 17.0 min. We therefore suggest this molecule consists of a methylated dicyclic aromatic with a N-heterocycle.

**Table 1 | Seven confirmed molecular identities in SAM-Flight Model Gas Chromatograph 1 and Gas Chromatograph 2 data**

| Peak no. | Molecule | Abundance (nmol) | SAM-FM scaled Rt (min) | SAM GC Breadboard Rt (min) | GC column |
|---|---|---|---|---|---|
| 1 | Trimethylbenzene (*m/z* 120) | 1.7 ± 0.3 | 13.5 | 12.9 | GC1 |
| 2 | Tetramethylbenzene (*m/z* 134) | 0.6 ± 0.1 | 15.3 | 15.6 | GC1 |
| 11 | Methyl benzoate (*m/z* 105) | 0.7 ± 0.1 | 9.5 | 9.2 | GC2 |
| 15 | Dihydronaphthalene (*m/z* 130) | 0.2 ± 0.0 | 12.2 | 12.7 | GC2 |
| 7 | Naphthalene (*m/z* 128) | 0.5 ± 0.1 | 17.5 | 17.5 | GC1 |
| 7 | Naphthalene (*m/z* 128) | 0.3 ± 0.0 | 13.1 | 13.7 | GC2 |
| 8 | Benzothiophene (*m/z* 134) | 0.2 ± 0.0 | 17.9 | 17.9 | GC1 |
| 8 | Benzothiophene (*m/z* 134) | 0.4 ± 0.1 | 13.4 | 13.7 | GC2 |
| 21 | Methylnaphthalene (*m/z* 142) | 0.1 ± 0.0 | 15.7 | 17.0 | GC2 |

Additional data include complementary peak number on chromatograms in Figs. 1 and 2, molecular ion, abundance in nmol, SAM-Flight Model retention time scaled to each channel, SAM-gas chromatograph-mass spectrometer breadboard retention time for comparison with candidate molecules, and which column the detection was on.

Highly volatile molecules detected in GC1 separations were likely lost from GC2, which warmed as adjacent GC1 heated, volatilizing some of the passively retained analytes at GC2's front end. Short-chained carboxylic acids likely passed through GC2 undetected before the start of the GC2 measurement. The carbon chains of saturated carboxylic acid methyl esters (CAMEs) that should elute within the time range of GC2 are $C_4$ through $C_{11}$; $C_4$ through -$C_{16}$ would theoretically elute in the GC1 range. No CAMEs, including the $C_9$ internal standard, were detected in the TMAH experiment.

## Complimentary laboratory experiments

As a geochemical comparison, Murchison meteorite and other carbonaceous chondrites can be used as a reference for exogenous organic matter delivered to Mars[23]. To help interpret SAM data and the potential origin for organics detected, benchtop laboratory experiments were performed on the Murchison meteorite. During TMAH thermochemolysis, the Murchison meteorite insoluble organic matter (IOM)[24] is degraded into its constituent parts, some of which are detectable with GC-MS[5,23]. These benchtop experiments included neat pyrolysis and TMAH thermochemolysis with and without a small contribution of MTBSTFA vapor to mimic the MTBSTA leak in SAM. Experiments were performed with both flash pyrolysis at 600 °C and the SAM pyrolysis ramp of 35 °C/min.

These benchtop experiments on the Murchison meteorite liberated a variety of aromatic acids and hydrocarbons, aliphatic carboxylic acids, and organosulfur compounds. Sixteen of the 28 species confirmed or tentatively identified in the SAM-FM TMAH experiment are also identified in the TMAH thermochemolysis of the Murchison meteorite. Molecules detected with neat pyrolysis and/or TMAH thermochemolysis of Murchison include benzene, toluene, tri-, tetra-, and pentamethylbenzene, trimethylbenzenamine, ethyltetramethylcyclopentadiene(?), naphthalene, methyl benzoate, benzothiophene, methylnaphthalene, dimethyl-indole, methoxybenzene, and methoxynaphthalene[23,25] (Table S-1).

These benchtop experiments demonstrated that, with flash TMAH thermochemolysis at 600 °C, aromatic acids are generated from cleaved ester aliphatic linkages joining the IOM macromolecular framework in the Murchison meteorite[24]. These acids could then be further decarboxylated under the pyrolysis conditions, yielding benzene and naphthalene. However, under SAM-like pyrolysis conditions, those aromatic acids (e.g., benzoic acid) that might have decarboxylated into an alkylbenzene would instead be methylated into volatile, thermally stable methyl ester derivatives[24]. Both products (e.g., aromatic acids and their decarboxylated alkyl-aromatic derivatives) are detected in the SAM-FM TMAH experiment.

Benchtop TMAH experiments yielded methylated and non-methylated species, indicating partial methylation of compounds from decreased TMAH thermochemolysis reaction rates at the slower SAM pyrolysis ramp. This finding is expected as TMAH derivatives from the SAM-like ramp experiment displayed reduced relative abundances, ca. two orders of magnitude lower as compared to flash conditions. Because of this, chromatograms from benchtop SAM-like TMAH thermochemolysis experiments contain a combination of peaks representing compounds detected in pyrolysis experiments with and without TMAH. Methylation of free benzene or naphthalene with TMAH is unlikely, as the reagent hydrolyzes labile H from polar functional groups which are lacking on these aromatic molecules. Direct C-methylation of naphthalene with TMAH was also not observed in benchtop experiments[26]. These results indicate that the methylated benzene and naphthalene derivatives in the SAM-FM TMAH experiment are derived from a macromolecular source like the Murchison macromolecular carbon.

## Detection of methyl benzoate

Methyl benzoate was detected in the SAM-FM TMAH experiment in GC2. Benzoic acid has been detected previously with SAM from both Mars-indigenous[3,7,27] and SAM-internal[7,21] sources. SAM-internal sources are detailed in refs. 7,19. Previous work has documented that benzoic acid may be produced through the oxidation of organic molecules present in the presence of perchlorates in the martian regolith and/or the thermal decomposition of the Tenax TA adsorbent present in the SAM trap(s). Although reactions of TMAH with 10 wt% $CaClO_4$ during benchtop flash pyrolysis experiments[28] have been shown to generate methyl benzoate amongst other products (but not naphthalene), this is not a potential source for the SAM-FM TMAH experiment. Perchlorate was not detected in the Mary Anning target, and regardless, average perchlorate concentrations in Martian regolith are far lower at 0.5 to 1.0 wt%. Benzoic acid and naphthalene have also been detected after overheating (>500 °C) and repeated activations of Tenax during benchtop experiments[21]. However, the SAM-FM Tenax has not been overheated.

Benzoic acid was postulated to be the preferred precursor for the chlorobenzene that was detected by SAM and demonstrated to be indigenous in the Cumberland mudstone at the base of Aeolis Mons[3,27]. Notably, chlorobenzene was detected at background levels in the Mary Anning neat pyrolysis experiment[6]. Chlorobenzene can be produced by reactions between benzoic acid and perchlorate at elevated temperatures[27], but again, perchlorates were not detected at Mary Anning. A portion of the benzoic acid derivatives generated in the TMAH and MTBSTFA[6] experiments at Mary Anning may still be related to the decomposition of Tenax in the HC trap. However, its non-detection in the neat pyrolysis of MA, where similar analytical conditions were applied, indicates at least a partial martian origin for benzoic acid.

The Murchison meteorite benchtop experiments demonstrated that pyrolysis-only treatments could generate benzoic acid, but only

with TMAH thermochemolysis does methyl benzoate form, even without the use of a hydrocarbon trap[23]. The presence of methyl benzoate in the Mary Anning TMAH experiment confirms that TMAH successfully reacted with the Martian sample to generate a CAME.

## Detection of benzothiophene

A variety of sulfur-bearing molecules have been previously detected on Mars with SAM (i.e.,[3,5,6]), including dithiapentane, dithiolane, and tri-thiane, which were detected in the neat pyrolysis GC-MS analysis of Mary Anning. The benzothiophene detection reported here is robust, detected in both GC1 and GC2, and the first confirmation of ben-zothiophene on Mars, as the molecule was only weakly indicated in lacustrine sediments in the Gale crater floor[5], and was lacking from the MA neat pyrolysis experiment. These S-bearing molecules were not methylated but could have been liberated from a macromolecular source by TMAH thermochemolysis. No known pyrolysis or thermo-chemolysis process on SAM would generate benzothiophene as a SAM-byproduct, but it is prevalent in carbonaceous meteorites (see ref.[29] and references therein) and readily liberated by pyrolysis and TMAH thermochemolysis from carbonaceous meteorites such as Murchison[23], as well as found in-situ and confirmed by extraction in the Tissint meteorite[30,31]. Sulfurization has been invoked as a preservation mechanism for other S-bearing organics identified on Mars[5]. Together, these findings suggest that the benzothiophene in Mary Anning was liberated from an indigenous macromolecular source.

## Detection of methylated cyclic molecules

Methylated benzene and naphthalene products were detected, which suggests that a larger macromolecular structure was broken apart by the TMAH thermochemolysis process. The methylated benzene and naphthalene derivatives are potentially sourced from partial methy-lation of aromatic acids such as benzoic and naphthoic acids. Although $C_3$ and $C_4$ methylated benzene and $C_1$ and $C_2$ methylated naphthalene derivatives were generated with benchtop experiments on Murchison meteorite with SAM-ramp neat pyrolysis and TMAH thermo-chemolysis, methylated benzene and naphthalene derivatives were not generated in the Mary Anning neat pyrolysis experiment[6]. We note that none of the organic molecules identified in the TMAH thermo-chemolysis experiment at Mary Anning were also present in the Mary Anning neat pyrolysis experiment, indicating unique experimental conditions that yielded different organic molecules between the two experiments. There are more similarities between the Mary Anning MTBSTFA experiment and the TMAH experiment than there are between the TMAH experiment and the neat pyrolysis experiment, including the presence (or presence at background levels) of benzoic acid (in the t-BDMS or methyl ester form), diphenylmethane/methyl-biphenol (considered a SAM internal source), and naphthalene (potentially indigenous)[6].

## Detection of nitrogen-bearing molecules

The mass spectra for peak 22 in the TMAH experiment are most con-sistent with the N-heterocycle dimethyl-indole. The most similar molecules identified in benchtop TMAH thermochemolysis experi-ments with Murchison meteorite yielded di-, tri- and tetramethyl-indole, and 1-methyl-pyrrole[23]. Though several N-heterocycles have been identified in carbonaceous chondrites after aqueous or acid extraction[32–35] the same cannot be said for the martian meteorites, where non-terrestrial amino acids were detected in RBT 04262. N-heterocycles were not detected by the same technique in ALH84001, ALHA77005, EETA79001, or MIL03346[36], but have been confirmed in Tissint, Nakhla, and NWA1950[30]. We suggest that the molecule present at peak 22 consists of a methylated double-ring aromatic with a N-heterocycle. This is an exciting possibility, as N-heterocycles are fundamental components of astrobiologically relevant molecules, such as nucleic acids.

Molecules containing nitrogen heteroatoms were also detected in GC1 and/or GC2 data at peaks 9, 10, and 17 (potentially trimethyl-benzenamine, dimethyl-benzenamine, and tetramethyl-benzenamine, respectively)[37]. Similar molecules such as N,N,2-trimethyl-benzena-mine and N,N-dimethyl-benzenemethanamine are identified in the benchtop neat pyrolysis and TMAH experiments of the Murchison meteorite[23,38]. Several TMAH thermochemolysis products in these benchtop experiments did yield other molecules bearing dimethyla-mine functional groups, such as methyl 3-dimethylaminopropionate and N,N,N',N'-tetramethyl-1,2-ethanediamine[23]. Additionally, aliphatic amines have been detected in hydrolyzed water extracts of Murchison[39–41]. Furthermore, nitrogen functionality and the presence of aliphatic and aromatic nitrogen species have been determined from the Tissint meteorite[31].

Benzonitrile is another common pyrolysis product of some meteorites, and has been found in the Murchison meteorite[23] and in the martian meteorite Nakhla[40] with its Martian origin confirmed[30]. Benzonitrile was identified in the Mary Anning MTBSTFA experiment, and in other Glen Torridon drill targets (Groken and Nontron)[6], but was not identified from the SAM TMAH experiment. The three peaks associated with N-bearing organics indicated in this work may yet be products of reactions between TMAH and a macromolecular precursor in Mary Anning, although reactions between TMAH and the SAM hydrocarbon trap cannot be totally discounted. TMAH produces a variety of methylated amine products due to the methylamine com-ponent of the organic salt. During thermochemolysis, intermediate products of both TMAH and sample organics could react to form a greater variety of N-bearing products.

## Detection of oxygen-bearing molecules

Oxygen-bearing molecules indicated in the SAM-FM TMAH experi-ment are comparable to those identified in benchtop TMAH thermo-chemolysis of Murchison, such as methoxymethyl-benzene and anisole[23]. Phenol is the most abundant O-containing organic com-pound liberated from the Murchison meteorite using neat pyrolysis. With TMAH thermochemolysis (under 600 °C flash benchtop condi-tions), phenol is converted to anisole (methoxybenzene). Under SAM ramp TMAH conditions, both phenol and anisole are detected. In the SAM-FM TMAH data, it can be challenging to differentiate between these similar molecules. For example, peaks 12 and 16 are likely "a benzene ring with alcohol" and both species are reflected in the closest NIST matches (Figs. S11 and S15). Peak 24 is likely a "methoxy-bearing double ring aromatic" and the closest NIST match is methox-ynaphthalene (which was also identified in the Murchison TMAH experiments, 23). Peak 18 is likely "a single ring aromatic with alcohol and/or methyl functional groups" and the closest NIST match is tetramethyl-phenol. The detection of both species is consistent with the partial methylation of likely meteoritic macromolecular carbon due to decreased TMAH thermochemolysis reaction rates at the slower SAM pyrolysis ramp. The observation that more of these species were observed in the newly commissioned GC2 column, which has a sta-tionary film with lower (5%) phenyl moieties than GC1 (20%), supports the conclusion that these species are not artifacts from column bleed. Column degradation should yield at most a minor contribution to the molecules identified from this column, although hydrocarbon trap products may still be a partial source of the benzoic acid (see Sup-plemental Information) discussed above.

## Evolved gas analysis

All molecules detected in the SAM-FM TMAH experiment are cyclic. EGA results for select $m/z$ values are consistent with the presence of benzene, toluene, tri- and tetramethylbenzene, naphthalene, and methylnaphthalene (Table S-1), and these are consistent with species identified in benchtop TMAH thermochemolysis experiments of the Murchison meteorite[23]. Molecular signals in EGA cannot represent

components produced on the SAM traps as the gas flow is diverted directly into the MS without encountering the HC trap. Benzene, toluene, alkylbenzenes and traces of naphthalene and benzothiophene signals above blanks levels were present in SAM-FM EGA of pyrolysis products from ancient Martian lacustrine sediments at the base of Aeolis Mons[5]. The study here now confirms the detection of naphthalene and benzothiophene with GC-MS.

## Lack of aliphatic carboxylic acids

One class of molecules commonly liberated from carbonaceous meteorites via TMAH thermochemolysis is CAMEs. TMAH methylates carboxylic acids with high efficiency[42] and CAMEs have been generated from benchtop TMAH thermochemolysis of Murchison meteorite[23]. Benzoic acid in the form of methyl benzoate was the only carboxylic acid detected in the SAM-FM TMAH experiment. The lack of aliphatic CAMEs in these data suggests that (1) CAMEs are not present in detectable amounts with SAM and/or (2) flight operating conditions require optimization before they can be definitively detected, with the latter a preferred explanation as described above.

## Mars-indigenous sources of organics

This study documents the first in situ TMAH wet chemistry experiment on Mars, which was performed by the SAM-FM instrument aboard the Curiosity rover. The TMAH experiment was successful based on confirmed puncture of the foils on the TMAH cup, detection of the recovery standard 1-fluoronaphthalene, and presence of the TMAH decomposition product trimethylamine. The detection of methyl benzoate confirms that TMAH reacted to form this methyl ester product. This study has identified over 20 aromatic and cyclic molecules with methyl and ester/carboxylic acid functional groups, and sulfur-, oxygen-, and nitrogen-bearing organics, including the first detection of a possible N-heterocycle. Both one- and two-ring aromatics have been identified, including the first confirmation of both naphthalene and benzothiophene. Benzothiophene is a known component of meteoritic macromolecular carbon and represents the largest confirmed underivatized aromatic molecule identified thus far as indigenous to Mars. Molecular abundances in the SAM TMAH experiment range from $0.1 \pm 0.0$ to $1.7 \pm 0.3$ nmol, consistent with the range of abundances of individual molecules identified by SAM from other Gale crater outcrops[3,6]. The detection of single and dicyclic aromatics and macromolecular carbon in near-surface Mars outcrops is consistent with the findings from analyzes of martian meteorites[30,37,43] and the NASA Mars 2020 Perseverance rover[44-46]. The detection of 1–2 ring aromatics in the SAM TMAH experiment, but not in neat pyrolysis, suggests that these molecules derive from a macromolecular source that was cleaved by TMAH thermochemolysis.

We propose that this suite of organics represents TMAH thermochemolysis breakdown products from ancient organic macromolecular material that has been preserved in billions-of-years-old sedimentary rocks in Gale crater. Analysis of the spatial distribution of the organic matter is not possible with SAM, therefore the origin of this material as being introduced from meteorites, abiotically produced via aqueous processing such as serpentinization or electrochemical production, is currently unknown[22,30]. Regardless, confirmation of macromolecular organic matter supports the possibility that future optimized TMAH thermochemolysis experiments can liberate ancient biosignatures preserved in macromolecules on Mars (if present). The broad structural variety of organic molecules observed in situ from surface materials suggests some chemical diversity is preserved in ancient Martian sediments despite >3.5 billion years of diagenesis and radiation exposure. These results expand the library of confirmed and suggested organic molecules preserved over deep geologic time in the Martian near-surface and confirm the presence of macromolecular carbon on Mars.

## Methods

### Mary Anning sample parameters and context

Details of the stratigraphy and geological context are presented elsewhere (see ref.[6] and references therein). In brief, the Mary Anning target is on a bedrock outcrop called Mozie_Law, located within the Knockfarrill Hill member of the Carolyn Shoemaker Formation within the Glen Torridon strata. The Glen Torridon region was previously informally named the "phyllosilicate trough" and is part of the Mt. Sharp group of lacustrine-related sediments. Three members from two formations of the Mt. Sharp group are exposed in Glen Torridon - they are (in stratigraphic order) the Jura member, the Knockfarrill Hill member, and the Glasgow member. The Jura member is the uppermost unit of the Murray formation and consists of mudstones and fine sandstones deposited in a low-energy environment. The transition from Jura to Knockfarrill Hill is interpreted as a change to a nearer-shoreline fluvial-influenced environment[12,47,48], with the Knockfarrill Hill and Glasgow members belonging to the Carolyn Shoemaker formation. The Mozie_Law outcrop is a finely laminated and cross-bedded sandstone.

Multiple drill holes were needed to obtain sufficient sample quantities for all the desired experiments, so a second drill was acquired close to the original drill site (MA1) and called Mary Anning 3 (the Mary Anning 2 site was assessed but not selected for drilling)[11]. An EGA was first performed, followed by a neat pyrolysis (without derivatization agents) standard SAM-GC-MS experiment with MA1 sample. Following that, MA3 sample was drilled and used in the first TMAH experiment and then a MTBSTFA wet chemistry experiment.

Curiosity drilled the Mary Anning 3 target of the Knockfarrill Hill member and, on sol 2879, delivered six portions of drilled fines to a SAM-FM cup containing 500 μL of TMAH reagent and two recovery standards (34 nmol 1-fluoronaphthalene and 25 nmol pyrene; via the Sample Manipulation System (SMS). Nonanoic acid (-12.5 nmol) was also present as an internal standard for the reaction[2,6]. The sample soaked in TMAH for 19 min at -25 °C (SAM ambient) under He flow (0.8 sccm, 25 mbar) before heating at 35 °C.min$^{-1}$ to -550 °C. Volatile products were split into two measurements: EGA and GC-MS. EGA continuously and directly measured bulk gas composition during heating with the mass spectrometer. For GC-MS analysis, evolved gas was periodically subsampled by collection on the hydrocarbon trap, enabling most solvent and TMAH decomposition byproducts to vent to Mars and minimizing their co-elution with analytes of interest. Trapped analytes were then thermally released from the trap and the flow split to two GC columns (GC1 and GC2), run consecutively. Unlike other GC channels in SAM (e.g., GC4-6), GC1 and GC2 do not have injection traps at the front of the column. Therefore, the effective start of the retention time (Rt) for GC1 is when the hydrocarbon trap is desorbed, and for GC2 is 23 min after the HC trap is reopened, after GC1 analysis is completed.

### Mary Anning mineralogy

The mineralogy of the Mary Anning samples established by CheMin includes feldspar, clay minerals, Ca-sulfate, quartz, hematite, and Fe-carbonate (likely trace siderite)[6,13]. Trace and/or amorphous Fe sulfur (likely sulfate) phases were inferred to be present in MA[49,50]. The gases released during SAM EGA (from MA2) included $H_2O$, $CO_2$, $SO_2$, $H_2$, HCl, $H_2S$, and CO. $O_2$ and NO were not detected. Some of the $H_2O$ evolved from MA could result from the bassanite detected by CheMin[13]. $H_2O$ evolved between -350 and 550 °C, as well as the small $H_2O$ peak at -820 °C, can be attributed to the clay minerals in MA. The MA sample evolved a sharp $CO_2$ peak, and $CO_2$ evolved above ~400 °C in EGA data can be attributed to siderite decomposition, which was detected in MA with CheMin. Thermal decomposition of oxidized organic molecules (e.g., oxalates[51-53]), and oxidation of reduced organics (SAM background or indigenous to the sample) during heating, also possibly contributed $CO_2$ at a range of temperatures, including in the

200 °C–550 °C range. The lack of NO evolved pointed to absent nitrate/nitrite salts. The nature of SAM HCl EGA curve suggested the presence of trace chloride salts.

## SAM derivatization and thermochemolysis experiments

The SAM SMS contains 74 cups, nine of which are foil-sealed Inconel metal cups for derivatization. Seven of the nine derivatization cups contain a mixture of MTBSTFA and DMF for derivatization, and the two others are filled with strongly alkaline (pH 12) TMAH in methanol for thermochemolysis/methylation[2,54] (Fig. S-27). Each thermochemolysis cup contains two separate reservoirs. The outer volume is 0.5 ml of a TMAH in methanol (25%) with 25 nmol of pyrene and 34 nmol 1-fluoronaphthalene (which do not react with TMAH). The inner volume contains ca. 12.5 nmol nonanoic acid, which had been sealed under vacuum inside a separate foil-capped reservoir[2]. Nonanoic acid is used to determine the methylation efficiency of the TMAH reaction of carboxylic acid after the two metal foils are punctured. The TMAH thermochemolysis experiment on SAM utilized elements of the standard EGA/GCMS sequence. The 900 °C cup preconditioning step cannot be used with the sealed liquid-containing metal cups, and the thermochemolysis experiment needed to occur in the same sol. The experimental procedure combined the manifold conditioning at 135 °C, cup puncture, sample delivery and EGA/GCMS analysis into a single sequence of experiments to minimize the time between sample delivery and pyrolysis and minimize solvent loss during extraction as detailed below.

The experiment was divided into the following steps: an initial pump down and conditioning of the sample manifold, a cup puncture with sample drop-off, then pyrolysis EGA and finally GCMS. Following the conditioning of the manifolds, QMS background scans were conducted before sample introduction. The sample drop-off moved the chosen sealed cup to the puncture station in the SMS and punctured both foils before moving the cup to the inlet for sample acceptance. The rover delivered six portions, ca. 163 ± 62 mg, to the cup which was then moved and sealed in the SAM pyrolysis oven under ~0.8 sccm helium flow.

Sample mass is estimated at ca. 163 mg due to the specific drilling approach used to obtain and deliver samples to SAM. The drilling approach is termed feed-extended sample transfer (FEST). With FEST, drilled sample fines were stored in the drill stem, and this was vibrated and rotated to deliver subsamples of drilled fines to SAM cups through inlets in the rover deck. Sample fines from MA were delivered to cups in SAM's Oven #1 ring. A combination of CheMin-derived sample mineralogy and processing of SAM EGA data was utilized to estimate the sample mass that SAM received from FEST delivery for samples from MA1 and MA2. This approach was detailed in McAdam et al. (2020)[55], but in summary, delivered sample mass estimates were based on abundances of minerals detected by CheMin that also were expected to release gases during SAM pyrolysis (i.e., phyllosilicates or hydrated Ca sulfates such as bassanite or gypsum, which release $H_2O$ during pyrolysis). This was carried out by deconvolving SAM EGA traces, in this case $H_2O$ traces, into several peaks. Peaks at temperatures consistent with thermal decomposition of phyllosilicates or hydrated Ca sulfates were ascribed to those phases. The amount of $H_2O$ in peaks attributed to a given mineral was assumed to derive from the wt. % of that mineral obtained from CheMin data. Determination of the moles of $H_2O$ evolved in a given mineral's EGA peak and the moles of $H_2O$ expected from a given wt. % of phyllosilicate or hydrated Ca sulfate based on CheMin data was used together to estimate the mass of the sample analyzed by SAM[13]. MA1 mass was estimated at 113 ± 43 mg from 4 portions from FEST delivery, and MA2 mass was estimated at 105 ± 40 mg from 4 portions from FEST delivery. Based on averages of these calculations, ca. 27.25 ± 10.375 mg were delivered per portion. The TMAH experiment used 6 portions of sample drilled from MA3. Therefore, we estimate that the mass delivered was ca. 163 ± 62 mg.

In a standard sequence without a derivatization or thermochemolysis cup, an empty quartz cup is preconditioned under helium flow in the pyrolysis oven to ~900 °C and then allowed to cool to ambient temperature before delivery of sample portions to the cleaned cup, followed by pyrolysis heating for EGA and GC-MS analyses. The gases released in a chosen range of oven temperatures, called temperature cuts, are sent to the SAM HC trap for subsequent GCMS analysis. Without derivatization or thermochemolysis, an EGA/GCMS sequence takes two or three sols whereas thermochemolysis and derivatization are generally performed in one sol. The thermochemolysis experiment utilized a linear 35 °C min⁻¹ ramp of the punctured cup containing the fluids and the delivered sample up to ~550 °C under He flow. Two techniques were employed to help to mitigate oversaturation of the SAM HC trap with TMAH in methanol during the temperature range below 250 °C, where the majority of the MTBSTFA vapor is expected. First, the SAM HC trap was kept at ca. 85 °C, a temperature that efficiently traps organics while allowing TMAH to pass through. Second, although EGA continuously and directly measured bulk gas composition during heating with the mass spectrometer, a portion of the evolved gas from the same sample was trapped and analyzed by GC-MS for molecular identifications, with the remaining portion vented to Mars's atmosphere via an exhaust line. During pyrolysis heating, portions of the evolved gas flow were directed to the hydrocarbon trap and subsampled at the cadence of 5 s for every 60 s from ambient to 300 °C, then 20 s for every 40 s from 300 °C to 550 °C. The subsampling was to allow the large solvent and TMAH decomposition byproducts to vent to Mars and not overwhelm the hydrocarbon trap (effectively serving as a split injection). The 550 °C oven temperature was empirically determined to be the ideal max temperature for TMAH thermochemolysis[53]. When the cup reached its maximum temperature, the HC trap was cooled to ca. 20 °C at a ~9 °C min⁻¹ rate to effectively collect both derivatized and non-derivatized analytes. After reaching the maximum cup temperature, the manifold was pressurized to 900 mbar, and the HC was heated to 320 °C in 5 min and held at the maximum temperature for 3 min to release the collected gases onto two GC columns for GCMS analysis.

## Gas chromatography

These evolved gases were then released in a dual column experiment to two of the six SAM GC columns: GC1–MXT-20 (polydimethylsiloxane +20% phenyl; 30 m/0.25 mm/0.25 μm) and GC2–MXT 5 (polydimethylsiloxane +5% phenyl; 30 m/0.25 mm/0.25 μm) channels of the chromatograph. GC analysis with GC1 was performed first while the GC2 channel was kept at the lowest possible temperature to keep the sample focused on the channel inlet. The analysis with GC2 began after the GC1 analysis was complete. Both columns are 30 m in length, with a 0.25 mm internal diameter and 0.25 μm film thickness, and are designed to analyze and separate low- to medium-molecular-weight mid-polar organics from 5 to 15 carbon atoms, under the operating conditions used for SAM. Neither column contains an injection trap (IT), commonly used to retain and then release analytes from the hydrocarbon trap onto the column. As mentioned in the main text, GC4 with its IT was intended to be the second column in this dual-column experiment but clogged during the first GC-MS analysis and could not be used for the TMAH experiment. Thus, we elected to commission GC2 for this experiment, although it does not utilize an injection trap.

During the time of pyrolysis, a portion of the gases released from the sample was sent to the HC trap (ca. 85 °C) in preparation for the GCMS analysis, while a small percentage went directly to the MS for the EGA (Fig. S-28). The HC trap was then heated to 310 °C and flushed with helium to release the molecules into both columns. The analytes released from the HC trap went directly into the MXT-20 column, immediately starting the analysis. The MXT-20 column was heated to 250 °C at a rate of 10 °C·min⁻¹. During the GC analysis with the MXT-20

column, the remaining analytes were sent to the MXT-5 column. Upon completion of the analysis with the MXT-20 column, the MXT-5 column was heated from ca. 60 °C (held for 60 min) to 230 °C (but ended heating prematurely at 190 °C) using a 10 °C min$^{-1}$ ramp rate. The exact split ratio between the two columns is unknown. The helium flow rate during the experiment was regulated at ~0.8 sccm. In summary, the timeline of the entire run included the EGA, followed by the MXT-20 analysis and ending with the MXT-5 analysis.

### MA follow-up analyses

After the analysis of MA, both MTBSTFA and a clean-up GCMS analysis were performed. The MTBSTFA analysis is detailed in ref.[6]. The clean-up GCMS analysis method was a standard HC trap and MXT-20/MXT-5 cleanup. Follow-up analyzes were implemented from the beginning of the mission to clean the transfer lines, traps, and columns from eventual condensed/trapped organics and inorganics that would not have come through in the original sample. They also minimized the carry-over of reagents or products to subsequent experiments.

### Abundance calculations

To calculate the abundances of trimethylbenzene, tetramethylbenzene, benzoic acid methyl ester, dihydronaphthalene, naphthalene, benzothiophene, and methylnaphthalene from the flight chromatograms, the major ion mass fragments of $m/z$ 120, $m/z$ 134, $m/z$ 105, $m/z$ 130, $m/z$ 128, $m/z$ 134, and $m/z$ 142, respectively—were fitted with Gaussian curves using the Igor Pro 8 software (WaveMetrics) and the areas under the curves were measured. The peak areas of each individual $m/z$ value contributing to the mass spectra of trimethylbenzene ($m/z$ 105 (base peak), 120, 119, 77, 106, 91, 39, 79, 121, 51), tetramethylbenzene ($m/z$ 119 (base peak), 134, 91, 133, 120, 135, 77, 117, 39, 41), benzoic acid methyl ester ($m/z$ 105 (base peak), 77, 136, 51, 50, 106, 78, 92, 76, 74), dihydronaphthalene ($m/z$ 130 (base peak), 129, 115, 128, 127, 131, 51, 64, 77, 63), naphthalene ($m/z$ 128 (base peak), 51, 63, 64, 75, 77, 102, 126, 127, 129), benzothiophene ($m/z$ 134 (base peak), 89, 90, 135, 67, 63, 136, 69, 108, 133), and methylnaphthalene ($m/z$ 142 (base peak), 141, 115, 143, 139, 63, 57, 71, 89, 116) were calculated from the relevant fragment ions using the expected ratios from the NIST MS reference library[17]. The isomers for trimethylbenzene and dihydronaphthalene on SAM are unknown so the ratios from 1,3,5-trimethylenzene and 1,2-dihydronaphthalene were chosen from NIST for the calculations. Abundances were then calculated by summing all $m/z$ values contributing to both peaks and comparing the total areas with the peak areas from five hexane GCMS measurements that were conducted on the SAM instrument during preflight calibrations[2]. The average electron ionization cross-section of hexane[56–59] of 19.7 Å$^2$ was used to evaluate the molar response difference between hexane and the six molecules considered here, and we calculated the ionization cross-section of these molecules using the bond contribution method described previously[56].

### Laboratory experiments

The laboratory experiments to support the interpretation of the flight results were performed on a GC Trace Ultra Chromatograph coupled to an ISQ LT MS from Thermo Fisher. Liquid (~0.1–0.5 μl) and gas (~0.5 ml) injections of the standard molecules were performed using a split/splitless injector in split mode with average split ratios between 1:50 and 1:100. The temperature of the injector and the ionization source were set at 250 °C and the GCMS transfer line at 300 °C to prevent condensation of the derivatized molecules in the instrument as they all have lower boiling points. The MS was set to scan the ions produced from the electron impact ionization source (electron energy of 70 eV) in the $m/z$ 10–535 range. The carrier gas was helium (Air Liquide, purity ≥99.9999%) to match the SAM GCMS experiment on Mars. The temperatures of the columns and carrier gas flow conditions used are detailed in the following section. A commercial CDS5100

pyroprobe was coupled to the GCMS. The various liquid reagents were injected onto glass wool packed in an organic-free quartz tube. The quartz tube was then inserted into the interface of the pyrolyzer, and the sample was flash pyrolyzed at 300 °C under helium flow after the sample had gone through the equivalent of the SAM HC.

### Comparison of laboratory and flight retention times

To confirm or rule out the identification of the derivatized and non-derivatized molecules in the SAM chromatograms, we compared the flight retention time with the laboratory retention time of equivalent molecules measured in SAM flight operating conditions (temperature and pressure) using a laboratory column (that is, the spare MXT-20 and MXT-5 columns). A flow restrictor was placed between the injector and the column to allow a SAM-like flow. To make sure that the flight and laboratory retention times were comparable, we adjusted the dead time in the laboratory to the dead time in flight by injecting non-retained molecules into both columns and adjusting the pressure in the laboratory until the two dead times were identical. This method has been used repeatedly in the past to confirm the presence of the chlorohydrocarbons and S-bearing molecules detected in the CB and Mojave samples[4,5]. It has also been used to build a database of retention times of molecules of astrobiological interest that may be present on Mars and detectable in future flight chromatograms. For the spare MXT-5 column, ~0.5 ml of air was injected to measure the dead time in the laboratory, and the laboratory GC pressure was adjusted until the dead time matched the flight dead time measured for the Chirasil-β Dex column in the SAM flight OG chromatogram. The program temperature was set to match the flight-temperature program from 40 °C (6.5 min hold) to 190 °C using a 10 °C min$^{-1}$ ramp.

The MXT-20 column does not include an injection trap, which makes comparison between the laboratory and flight retention times challenging. For this column, the laboratory GC pressure had to be adjusted as well as the length of the initial temperature plateau of the column (which is unknown given the absence of an IT flash time). To do so, several retention markers from molecules detected in the MXT-20 flight chromatogram were used, including SO$_2$ and BSW. The two molecules are by-products formed from the reaction between MTBSTFA and water. The pressure was adjusted using the SO$_2$ retention time, and the length of the initial temperature plateau of the MXT-20 column was adjusted by matching the retention times of the two by-products detected in all SAM and laboratory chromatograms including MTBSTFA. Minor retention time shifts remain due to several factors, including residual MTBSTFA-DMF in the column and the traps, saturation of TMA in the columns, the slight difference between the flight and laboratory ramp temperatures, and the manufacturing process of both stationary phases[18]. The final temperature program of MXT-20 was a 5 min hold at 31 °C, followed, as in SAM, by a ramp at 10 °C min$^{-1}$ to a final temperature of 250 °C.

The procedure to select standards for this laboratory work began by performing preliminary identifications of the molecules in the SAM-FM TMAH data based on mass spectra. Preliminary identifications were then made, and standards were selected for retention time confirmation experiments on the flight spare columns. The number of standards analyzed with these laboratory experiments was downselected based on the number of analyzes that could reasonably be completed with the flight spare columns, constraints on the laboratory facility access, and availability (and total cost) of the standards desired. Table S-1 details the standards that were analyzed as comparisons to suspected molecules identified from mass spectra of the SAM-FM TMAH experiment.

## Data availability

All mission data used in this paper are archived in the NASA Planetary Data System. The data generated in this study are provided in the

Supplementary Information file or are available in the Figshare database under accession codes https://doi.org/10.6084/m9.figshare.26084191, https://doi.org/10.6084/m9.figshare.26084188, https://doi.org/10.6084/m9.figshare.26084185, and https://doi.org/10.6084/m9.figshare.26084182 [https://figshare.com/s/c87fd8c014d4088ccff6].

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

## Acknowledgements

This work was enabled by a large and dedicated team of engineers and scientists that designed, built, and operate the Mars Science Laboratory mission. Support for this work comes from: NASA MSL Participating Scientist Program under Grant #80NSSC22K0651 (AJW), the NASA MSL Participating Scientist Program (CHH, VF), Center for Research and Exploration in Space Science and Technology (CRESST) II Cooperative Agreement (NASA Award Number 80GSFC21M0002 (AW, MM, RW, ST, LC, CF, JMTL), NASA-GSFC grant NNX17AJ68G (MM, SSJ), NASA's Planetary Science Division Internal Scientist Funding Program through the Fundamental Laboratory Research (FLaRe) work package at NASA Goddard Space Flight Center and the Goddard Center for Astrobiology (JPD), the NASA Postdoctoral Program (AM), the Carnegie Endowment (AS), the University of Florida University Scholars Online Award (JR), and CNES (MM, AB, DB). Part of this research was carried out at the Jet Propulsion Laboratory, California Institute of Technology, under a contract with the National Aeronautics and Space Administration (80NM0018D0004) (ARV). The authors thank the extended SAM and Curiosity teams for helpful comments which greatly improved the manuscript.

## Author contributions

Conceptualization: A.J.W., J.L.E., P.R.M., A.B., C.F., C.S. Methodology: A.J.W., J.L.E., M.M., R.W., S.T., C.M., P.R.M., A.B., J.P.D., C.F., D.P.G., C.S., A.R.V. Investigation: A.J.W, J.L.E., M.M., R.W., O.M., S.T., J.R., C.M., P.R.M., A.B.B, A.B., V.F., H.F., C.F., S.S.J., J.M.T.L, A.C.M., R.N.G., C.P., C.S. Visualization: A.J.W., M.M., C.F. Funding acquisition: A.J.W., C.M., P.R.M. Project administration: A.J.W., C.M., P.R.M., A.R.V. Supervision: A.J.W., C.M., P.R.M. Writing – original draft: A.J.W., M.M., R.W., C.F. Writing – review & editing: A.J.W., J.L.E., M.M., O.M., S.T., C.M., A.C.M., P.R.M., A.B.B., A.B., D.B., L.C., J.P.D, H.F., C.F., D.P.G., C.H.H., S.S.J, J.M.T.L., A.M., C.P., A.S., R.E.S., C.S., M.T.T., A.R.V.

## Competing interests

The authors declare no competing interests.
