## [Transparent Peer Review file · Nature Communications]

Diverse Organic Molecules on Mars Revealed by the First SAM TMAH Experiment

Corresponding Author: Dr Amy Williams

Version 0:

Reviewer comments:

Reviewer #1

(Remarks to the Author)

The manuscript was previously submitted to Nature.

The comments will only focus on responses of the authors and improvements of the manuscript.

It has been substantially revised and clarified. The last issue is still the lack of confirmation thanks to laboratory experiments. If additional data of standards are not available the authors could modulate the assumption of identification and justify this lack.

Despite this, major concerns have been addressed by the revisions and manuscript needs minor revision to be considered for publication in Nature communication .

Reviewer #2

(Remarks to the Author)

I have read the comments by Referees #1, #2, and #3 for the previous version of the manuscript, as well as the authors' responses for the comments. If my understanding is correct, the points in this manuscript, which the authors want to stress, are as follows;

- 1) This paper is the first result from TMAH thermochemolysis of Martian rock.
- 2) This paper shows that macromolecular organic material, which can be cleaved by TMAH thermochemolysis, is present on Mars.

I don't think that the authors need to mind of Referee #1's criticism on isotopic labelling, although he/she gives sharp comments. It is very important to maximize the results under the limited resources and other constraints on space explorations, and thus I understand the authors' arguments.

On the other hand, several questions from Referees #2 and #3 are the same or similar, and they seem to be the main issues for which the authors need to give convincing explanations.

I see that, the question behind the question from Referees #2 and #3 is, "Has the TMAH thermochemolysis experiment been really successful?" I am sorry if this is an annoying comment, but the authors' response is "premised" on that the TMAH thermochemolysis experiment on Mars was successful. The authors explained that the methylated benzene and naphthalene derivatives are potentially produced via the TMAH thermochemolysis, which is true and understandable. However, they have not answered the relevant question from Referee #3, "Aren't these pyrolysates released without TMAH reaction just at the same temperature as TMAH experiment?". As long as I know, methylated benzene and naphthalene are the typical products from a simple pyrolysis (without TMAH) of terrestrial kerogen.

Here, Referees #1's indication is recalled: The methylated nonanoic acid internal standard has not been seen, despite the release of TMAH and the nonanoic acid into the sample cup. In the manuscript, the reason for this is explained that the

sampling cadence led to the loss of the nonanoic acid. However, not only the internal std, any aliphatic carboxylic acid methyl esters from the Martian rock has not been seen, either. Isn't there possibility that the Martian samples and TMAH did not react sufficiently?

My question may have bias and misunderstanding, but, it is very important for the authors to show the irreplaceable evidence that Martian rock samples and TMAH reacted, for preventing any misunderstanding from reviewers, some members of the scientific community, and readers. Otherwise, the points 1 and 2 above are not very convincing.

Version 1:

Reviewer comments:

Reviewer #1

(Remarks to the Author)

Title : Diverse Organic Molecules on Mars Revealed by the SAM TMAH Experiment

Authors: Williams et al.

Analysis of organic molecules from Mars samples treated by thermochemolysis

The comments will only focus on improvements of the manuscript.

It has been substantially revised and clarified. The last issue being the lack of confirmation thanks to laboratory experiments is however better documented and the authors have smoothened the assumption of identification. SI has also being improved with all typos and wrong labelling corrected.

Major concerns have been addressed by the revisions and manuscript is suitable for publication in Nature communication .

Reviewer #2

(Remarks to the Author)

The authors answer clearly to each comments from the reviewer, and their explanation about the evidence that the TMAH thermochemolysis experiment was successful (detection of trimethyl amine, methyl benzoate) is convincing. Their previous studies on the benchtop experiments on Murchison meteorite and the on-site experiments at Mary Anning, Mars with both SAM-ramp neat pyrolysis and with TMAH thermochemolysis strongly help interpreting what the pyrolysate molecules are unique to TMAH experiment. These points have been well clarified in the first and last parts in the revised manuscripts. Addition of the paragraph (Lack of aliphatic carboxylic acids) will also answer readers' question clearly. The revised manuscript has demonstrated the authors' sincere efforts in accordance with the reviewer's concerns and has been very much improved. This version is acceptable for publication in Nature communication.

Thank you both for reviewing this manuscript yet again. We deeply appreciate the opportunity to address the issues brought up in your comments and feel that the edits we made based on those comments have greatly improved and clarified the manuscript. We have reorganized the discussion to highlight the confirmed molecular identifications and sources of those molecules. We very much look forward to having this work published and shared with the community.

Reviewer #1, (Reviewer 2 previously) (Remarks to the Author):

The comments will only focus on responses of the authors and improvements of the manuscript. It has been substantially revised and clarified. The last issue is still the lack of confirmation thanks to laboratory experiments. If additional data of standards are not available the authors could modulate the assumption of identification and justify this lack.

We have modified the text anywhere a laboratory experiment was not possible. As an example, we revised in lines 150 and 165 to make clear that those molecules without corresponding retention time experiment cannot be confirmed, but that we can speculate on their identities based on mass spectra.

We added a paragraph in the SI (under the section “Comparison of laboratory and flight retention times”) that details and justifies our constraints on standard acquisition and analysis for the retention time experiments.

We also created a table in the SI (Table S-1) that lists confirmed and potential molecule identities from the SAM-FM TMAH experiment and analytical standards analyzed with the SAM-BB and flight spare columns. Not every potential identity was compared to an analytical standard. The retention time (Rt) comparison between the molecule on SAM-FM and the SAM-BB is reported for those standards that were analyzed. Noted as Y for yes are those molecules that were also detected in the Murchison meteorite with SAM-like neat pyrolysis or TMAH thermochemolysis from Mojarro et al. (2023).

We ask that if there is a specific justification that you expected to see in these revisions and do not, please do communicate that to us and we will readily assess if we can include it!

Despite this, major concerns have been addressed by the revisions and manuscript needs minor revision to be considered for publication in Nature communication.

Thank you!

Reviewer #2 (Reviewer 3 previously) (Remarks to the Author):

I have read the comments by Referees #1, #2, and #3 for the previous version of the manuscript, as well as the authors’ responses for the comments. If my understanding is

correct, the points in this manuscript, which the authors want to stress, are as follows; 1) This paper is the first result from TMAH thermochemolysis of Martian rock. 2) This paper shows that macromolecular organic material, which can be cleaved by TMAH thermochemolysis, is present on Mars. I don't think that the authors need to mind Referee 1's criticism on isotopic labelling, although he/she gives sharp comments. It is very important to maximize the results under the limited resources and other constraints on space explorations, and thus I understand the authors' arguments.

Thank you!

On the other hand, several questions from Referees #2 and #3 are the same or similar, and they seem to be the main issues for which the authors need to give convincing explanations. I see that, the question behind the question from Referees #2 and #3 is, “Has the TMAH thermochemolysis experiment been really successful?” I am sorry if this is an annoying comment, but the authors' response is “premiered” on that the TMAH thermochemolysis experiment on Mars was successful. The authors explained that the methylated benzene and naphthalene derivatives are potentially produced via the TMAH thermochemolysis, which is true and understandable. However, they have not answered the relevant question from Referee #3, “Aren't these pyrolysates released without TMAH reaction just at the same temperature as TMAH experiment?”. As long as I know, methylated benzene and naphthalene are the typical products from a simple pyrolysis (without TMAH) of terrestrial kerogen.

With regards to the question “Has the TMAH thermochemolysis experiment been really successful?”, we can readily conclude that it was both from the engineering and scientific standpoint, as the thermochemolysis product of TMAH, trimethylamine (TMA), was abundantly identified in the EGA and GCMS data from the *in situ* experiment (see Figure 1). If TMAH was not released and decomposed under heating, TMA would not have been generated [Line 117]. The response to the question below provides more detail on this question, which we interpret as “How do we demonstrate that the Martian sample and TMAH reacted” (from the scientific standpoint).

With regards to the referee question “Aren't these pyrolysates (methylated benzene and naphthalene derivatives) released without TMAH reaction just at the same temperature as TMAH experiment?”, Mojarro et al. (2023) demonstrated that C₃ and C₄ methylated benzene, and C₁ and C₂ methylated naphthalene derivatives were generated with benchtop experiments on Murchison meteorite with both SAM-ramp neat pyrolysis and with TMAH thermochemolysis (Lines 284-288).

However, methylated benzene and naphthalene derivatives (beyond the known HC trap sources for biphenyl and methylbiphenyl/ diphenylmethane) were not generated in the Mary Anning neat pyrolysis experiment on Mars (Millan et al., 2022), despite the Mojarro et al. (2023) work

predicting their presence. We also note here and in the related response to the question below that none of the organic molecules identified in the TMAH thermochemolysis experiment at Mary Anning were also present in the Mary Anning neat pyrolysis experiment, indicating a unique experimental condition that yielded different organic molecules between the two experiments (**Lines 288-291**). There are more similarities between the Mary Anning MTBSTFA experiment and the TMAH experiment than there are between the TMAH experiment and the neat pyrolysis experiment, including the presence (or presence at background levels) of benzoic acid (in the *t*-BDMS or methyl ester form), diphenylmethane/methylbiphenol (considered a SAM internal source), and naphthalene (potentially indigenous). This is reported in Millan et al. (2022) and cited in our manuscript.

Your point that methylated benzene and naphthalene derivatives are the typical products from a simple pyrolysis (without TMAH) of terrestrial kerogen is a critical point that we have further reinforced in the manuscript - that these molecules are the typical products from terrestrial kerogen, which indicates that a Martian kerogen-like (or macromolecular carbon, as we name it in the manuscript) material is being pyrolyzed and/or thermally hydrolyzed and methylated in the experiment. We continue to highlight the point in the manuscript that this discovery alone is notable, that Martian macromolecular carbon is present and detectable within centimeters of the irradiated Martian surface.

Here, Referees #1's indication is recalled: The methylated nonanoic acid internal standard has not been seen, despite the release of TMAH and the nonanoic acid into the sample cup. In the manuscript, the reason for this is explained that the sampling cadence led to the loss of the nonanoic acid. However, not only the internal std, any aliphatic carboxylic acid methyl esters from the Martian rock has not been seen, either. Isn't there possibility that the Martian samples and TMAH did not react sufficiently?

Although aliphatic carboxylic acid methyl esters were not identified in the *in situ* experiment, the presence of methyl benzoate (benzoic acid methyl ester or BAME) is proof that the sample reacted with TMAH. Experiments performed in Mojarro et al. (2023) on Murchison meteorite treated with neat pyrolysis and TMAH thermochemolysis demonstrated that pyrolysis-only treatments could generate benzoic acid, but only with TMAH thermochemolysis did benzoic acid methyl ester form [**Further documented in lines 250-255**].

Additionally, three experiments were performed on this Mary Anning sample: neat pyrolysis, TMAH thermochemolysis, and MTBSTFA derivatization (as reported in Millan et al., 2022). Benzoic acid derivatives were detected in both Mary Anning wet chemistry analyses - TMAH and MTBSTFA (and for what it's worth, after the first full cup derivatization experiment on Mars on the Ogunquit Beach (OG) sample (Millan et al., 2021)), but neither benzoic acid nor its derivatives were detected in the Mary Anning neat pyrolysis experiment (**lines 260-264**). Therefore, we can

conclude that the wet chemistry experiments liberated and volatilized benzoic acid such that we detect it on SAM with wet chemistry experiments.

We comment in the text (**lines 241-259**) that benzoic acid may be produced through the oxidation of organic molecules present in the martian regolith and/or the thermal decomposition of the Tenax TA adsorbent present in the SAM trap(s) (Millan et al., 2021). Benzoic acid is a likely precursor of the chlorobenzene that has been detected on Mars in the Cumberland sample (Freissinet et al., 2020). Although a portion of benzoic acid may still be related to the decomposition of Tenax, its non-detection in MA and GE, where similar analytical conditions were applied, suggests at least a partial martian origin.

We also note that none of the organic molecules identified in the TMAH thermochemolysis experiment at Mary Anning were also present in the Mary Anning neat pyrolysis experiment, indicating a unique experimental condition that yielded different molecules between the two experiments (**lines 288-295**). As we noted in the response above, there are more similarities between the Mary Anning MTBSTFA experiment and the TMAH experiment than there are between the TMAH experiment and the neat pyrolysis experiment.

As it is relevant to this response, we also wish to respond to original Reviewer #1's comment (since you mentioned it): "The methylated nonanoic acid internal standard has not been seen, despite the release of TMAH and the nonanoic acid into the sample cup." As noted in the text and the Supplemental, the TMAH experimental cups consist of two sealed foil caps. The first outer foil contains the TMAH in methanol, 34 nmol of 1-fluoronaphthalene and 25 nmol of pyrene both as recovery standards that would not react under thermochemolysis. The second inner foil contained the 13 nmol of nonanoic acid intended as the internal standard. Our engineering telemetry data indicate that a full puncture of both foils occurred, and 1-fluoronaphthalene was readily detected in the Mary Anning experiment (we highlight this again in **lines 114-128**). We also explain in the text that the elution time for the pyrene was outside the time and energy constraints of the columns we used for this experiment, therefore we did not expect to see it, and we did not. Therefore, we expect that the sample interacted with this the TMAH in methanol, and all three standards were liberated as part of the experiment. As we mention in the manuscript, the lack of detected nonanoic acid or its methyl ester derivative is most likely due to the sampling cadence that sent small 'sniffs' to the HC trap and the GC columns to avoid saturation. The lack of carboxylic acid methyl ester detection from the sample may just mean that they were not present in high enough concentration to be detected given the sampling cadence we initially selected for the experiment. Indeed, macromolecular carbon would be expected to be far more resilient in the Martian near surface than free carboxylic acids, and our results are consistent with this interpretation – that macromolecular carbon was broken apart and components liberated and volatilized by TMAH thermochemolysis.

My question may have bias and misunderstanding, but, it is very important for the authors to show the irreplaceable evidence that Martian rock samples and TMAH reacted, for preventing any misunderstanding from reviewers, some members of the scientific community, and readers. Otherwise, the points 1 and 2 above are not very convincing.

This is a recap of the points made above. In summary, the Martian rock sample and TMAH must have reacted based on these lines of evidence:

1. Several molecules with sulfur heteroatoms were detected with neat pyrolysis at Mary Anning, but benzothiophene was readily detected at elevated abundance in the TMAH experiment only. The detection of benzothiophene is therefore aligned with the detection of other S heteroatoms and heterocycles detected at Mary Anning, and its liberation from the Martian sample is consistent with cleavage of Martian macromolecular material. This is the first confirmed detection of benzothiophene, with only hints of it from weak m/z profiles in the Mojave and Confidence Hills samples (Eigenbrode et al., 2018). Benzothiophene is a known component of meteoritic macromolecular carbon, and its detection with benchtop TMAH thermochemolysis from the Murchison meteorite is demonstrated in Mojarro et al. (2023).
2. TMAH thermochemolysis generates carboxylic acid methyl esters, and benzoic acid methyl ester was detected in the Mary Anning TMAH experiment and benzoic acid *t*-BDMS was detected in the Mary Anning MTBSTFA experiment. Benzoic acid was not identified in the Mary Anning neat pyrolysis experiment, suggesting that thermochemolysis and derivatization aided in cleaving a Martian macromolecular carbon source, with thermal hydrolysis and methylation generating the methyl ester form in the former example. Additionally, benzoic acid *t*-BDMS detected in the MTBSTFA experiment at Ogunquit Beach (OG, Millan et al., 2021) was interpreted to be at least partially Martian in origin. Therefore, the TMAH experiment reacted with benzoic acid that is at least partially Mars indigenous, and that was not detected in the neat pyrolysis experiment, to generate benzoic acid methyl ester.
3. None of the organic molecules identified in the TMAH thermochemolysis experiment at Mary Anning were also present in the Mary Anning neat pyrolysis experiment, indicating a unique experimental condition that yielded different organic molecules between the two experiments. There are more similarities in molecules generated between the Mary Anning MTBSTFA experiment and the TMAH experiment than there are between the TMAH experiment and the neat pyrolysis experiment.
4. Mojarro et al. (2023) demonstrated that C₃ and C₄ methylated benzene and C₁ and C₂ methylated naphthalene derivatives were generated with benchtop experiments on Murchison meteorite with SAM-ramp neat pyrolysis, TMAH thermochemolysis, and MTBSTFA derivatization. However, methylated benzene and naphthalene derivatives were only detected in the Mary Anning TMAH experiment, and neither in the Mary Anning neat pyrolysis experiment nor in the Mary Anning MTBSTFA experiment. Naphthalene was only possibly

detected in the MTBSTFA experiment (Millan et al., 2022). Regardless of source, the presence of these methylated derivatives indicates that macromolecular material was cleaved by TMAH thermochemolysis to generate these products.

5. TMAH decomposed (as expected) in the presence of Martian rock samples based on the generation of abundant trimethylamine (TMA) detected in EGA and both GC-MS columns.
6. The presence of the recovery standard 1-fluoronaphthalene in both GC column analyses confirms the TMAH cup foil was punctured and molecules from the reaction in the cup were flushed into the GC and the MS.